# Perirhinal cortex learns a predictive map of the task environment

David G. Lee[1,2,7], Caroline A. McLachlan[2,3,7], Ramon Nogueira [4,5], Osung Kwon[2,3], Alanna E. Carey[2,3], Garrett House[3], Gavin D. Lagani[3], Danielle LaMay[3], Stefano Fusi [4,5] & Jerry L. Chen [1,2,3,6] ✉

Goal-directed tasks involve acquiring an internal model, known as a predictive map, of relevant stimuli and associated outcomes to guide behavior. Here, we identified neural signatures of a predictive map of task behavior in perirhinal cortex (Prh). Mice learned to perform a tactile working memory task by classifying sequential whisker stimuli over multiple training stages. Chronic two-photon calcium imaging, population analysis, and computational modeling revealed that Prh encodes stimulus features as sensory prediction errors. Prh forms stable stimulus-outcome associations that can progressively be decoded earlier in the trial as training advances and that generalize as animals learn new contingencies. Stimulus-outcome associations are linked to prospective network activity encoding possible expected outcomes. This link is mediated by cholinergic signaling to guide task performance, demonstrated by acetylcholine imaging and systemic pharmacological perturbation. We propose that Prh combines error-driven and map-like properties to acquire a predictive map of learned task behavior.

The brain generates internal models of the environment that describe the relationship between stimuli, events, and outcomes. Models are learned through experience and can be stored as memories. These memories can be recalled to serve as predictions of upcoming stimuli or outcomes to guide ongoing task behavior. As sensory information is evaluated against internal models, they can generate at least two types of neural signals. Activity can report when sensory information does not match the prediction, referred to as a 'sensory prediction error'. Activity can also be reported when sensory information is predictive of an outcome such as a reward, referred to as a 'stimulus-outcome association'. In the sensory neocortex, sensory prediction errors are a hallmark of predictive coding, a proposed framework in which predictions of sensory information are generated and evaluated against actual sensory input[1–3]. Stimulus-outcome associations are the basis for cognitive maps in the hippocampus[4–6], a representation that reduces similar spatial and non-spatial associations to a lower-dimensional 'abstract' format[7,8]. This format is proposed to facilitate the generalization of novel stimulus-outcome associations[9,10].

The extent to which predictive coding and cognitive maps are aspects of distinct or common neurobiological processes is unclear. Recently, it has been proposed that the two theories could be considered part of a broader framework, referred to as a 'predictive map'[11,12]. During goal-directed sensory-guided behavior, sensory prediction errors and stimulus-outcome associations would both be readouts of a single predictive map of the task. This predictive map would be acquired and updated by a combination of error learning to minimize sensory prediction errors and associative learning to strengthen stimulus-outcome associations. The map would be used to predict upcoming task events and infer relationships between novel experiences. Different maps could be flexibly recalled depending on behavioral conditions.

[1]Department of Biomedical Engineering, Boston University, Boston, MA 02215, USA. [2]Center for Neurophotonics, Boston University, Boston, MA 02215, USA. [3]Department of Biology, Boston University, Boston, MA 02215, USA. [4]Center for Theoretical Neuroscience, Columbia University, New York, NY 10027, USA. [5]Department of Neuroscience, Columbia University, New York, NY 10027, USA. [6]Center for Systems Neuroscience, Boston University, Boston, MA 02215, USA. [7]These authors contributed equally: David G. Lee, Caroline A. McLachlan. ✉e-mail: jerry@chen-lab.org

To look for neural evidence of a predictive map, we focused on the perirhinal cortex (Prh), a zone of convergence between the sensory neocortex and the hippocampus[13–15]. Prh has multiple roles in sensory processing, including unitizing features, assigning relational meaning, signaling novelty, and temporal ordering of stimulus items[16–18]. These sensory- and memory-related functions suggest that Prh generates a model of relevant sensory information associated with task behavior. This suggests that functions associated with predictive coding and cognitive maps are combined and expressed in this area. Prh also receives dense cholinergic inputs[19–21]. Acetylcholine is involved in reward expectation and enhancing sensory processing related to predictive coding[22,23], as well as memory encoding and retrieval related to cognitive maps[24–26]. Cholinergic signaling could serve as a mechanism that would flexibly establish network states, enabling predictive maps to be recalled and utilized in Prh. Here, we investigated whether neural substrates in Prh support the acquisition, representation, and implementation of a predictive map of learned sensory-guided behavior.

## Results

### Evolution of sensory and motor variables during delayed non-match-to-sample task training

To investigate how predictive maps are acquired and updated, mice were trained to perform a delayed non-match-to-sample task that required them to classify sequentially presented whisker stimuli[27,28] (Fig. 1a). A motorized rotor was used to deflect multiple whiskers in either an anterior (A), or posterior (P) direction during an initial 'sample' and a later 'test' period. Mice were trained to report by go/no-go whether the presented sample and test stimuli were non-matching or matching. Within a trial, both deflections were also presented at one of two speeds ('fast' or 'slow'). Speed can be considered both a second stimulus dimension and a variation in the strength of the rotation direction. This means that animals need to consider relevant (direction) and irrelevant (speed) stimulus features in order to abstract a complex sensory relationship (non-match or match). Temporally dissociating the stimulus features into two distinct periods enabled us to investigate how predictive maps are evaluated when features are necessary but not yet sufficient to predict the outcome (sample) and when the combination of features can sufficiently predict the outcome (test).

Overall, training was divided into multiple training stages. Each stage was designed to assay aspects of stimulus-feature and stimulus-reward learning (Tables 1 and 2). The initial training stages consisted of one non-match stimulus condition (AP) and two match stimulus conditions (AA, PP). Training under these conditions was subdivided into 2 stages according to initial naive performance (T1) and learned performance (T2, $d' > 0.45$ for two consecutive sessions). Completion of T2 required the animal to unitize the sample and test stimuli and pair it with a reward. In the following stage (T3), the remaining held-out non-match condition (PA) was introduced, which required the animal to learn a new stimulus-reward contingency and generalize non-match and match across all possible combinations. Following successful learning of T3, delays between the sample and test stimuli were gradually extended up to 2 s (T4) to increase the temporal separation between the sample and test stimuli. During the final stage (T5), the rotor was fully retracted during the delay period to require animals to retain a working memory of the sample stimulus. This also prevented the animal from relying on potential positional cues that existed during T4 when the rotor remained in whisker contact throughout the delay period.

To investigate the role of Prh during task learning, we verified the anatomical coordinates of whisker-related Prh using reciprocal retrograde tracing between secondary somatosensory cortex (Supplementary Fig. 1). In initial experiments using a custom-built automated home cage training system to assay mice on a freely moving version of the task, we observed that chronic chemogenetic inactivation of Prh can decrease the rate of task learning (Supplementary Note S1, Supplementary Figs. 2–4). To study how population activity evolves in Prh with task learning, we performed chronic multi-depth two-photon calcium imaging in a cohort of head-fixed animals throughout the entirety of the training. A virus expressing the genetically encoded calcium indicator, RCaMP1.07 (AAV/PHP.eB-EF1α-RCaMP1.07), was delivered into Prh. To non-invasively image Prh using an upright two-photon microscope, a 2 mm microprism was laterally implanted to provide optical access along the cortical surface using a long working-distance objective (Fig. 1b). In addition to two-photon calcium imaging, high-speed videography was performed to measure whisker kinematics and whisking behavior (Fig. 1d, Supplementary Fig. 5).

We first asked whether animals changed their behavioral strategies with learning by measuring changes in sensory or motor variables. Unlike in other whisker-based sensory tasks[29,30], animals did not actively whisk during task performance. Whisking amplitude did not significantly change across training stages (Fig. 1e). We next compared whisker kinematics during different direction and speed conditions. Overall, whisker angle changes trended more in the anterior direction (Fig. 1f, sample, $P < 1 \times 10^{-5}$, $F_{4,593} = 8.01$, one-way ANOVA with post-hoc multiple comparison test; test, $P < 1 \times 10^{-4}$, $F_{4,592} = 7.10$, one-way ANOVA with post-hoc multiple comparison test). Despite this, posterior stimuli consistently produced more negative angle deflections than anterior stimuli. Posterior stimuli also consistently produced more negative curvature changes than anterior stimuli (Fig. 1g). Compared to fast conditions, slow conditions produced weaker negative angle deflections and curvature changes in the anterior direction. No difference was observed for either angle or curvature changes between slow and fast stimuli in the posterior direction. We additionally examined licking behavior across training. In early training stages, animals showed sporadic licking across different trial epochs, such as the sample and test period, but this became more restricted to the reporting period as animals advanced in the task (Fig. 1h, pre, $P < 1 \times 10^{-29}$, $F_{4,1159} = 38.8$; sample, $P < 1 \times 10^{-78}$, $F_{4,1159} = 109.0$; test, $P < 1 \times 10^{-43}$, $F_{4,1159} = 57.2$; report, $P < 1 \times 10^{-5}$, $F_{4,1159} = 8.3$, one-way ANOVA with post-hoc multiple comparison test).

### The perirhinal cortex learns sensory prediction errors

Given the specific changes in sensory and motor variables across learning, we sought to determine what aspects of sensory information are encoded in Prh. We analyzed calcium imaging data from 7 out of 9 animals that were successfully trained to T5 within ~60 training sessions (Fig. 1c). We focused on neural activity related to stimulus direction or speed and its relationship to task performance. Animals were primarily trained on directions with fast speeds (95% across T1–T4, 75% for T5), with slow speed trials provided as less frequent stimuli (5% across T1–T4, 25% for T5). Since whisker kinematic analysis shows that slower speeds produce fewer deflections in the anterior direction, weaker information about stimulus direction could affect task performance on slow-speed trials. Indeed, while animals were able to learn the task at fast and slow speeds, they performed worse on slow compared to fast speed conditions as they approached later training stages (T4, $P < 0.05$; T5, $P < 0.05$, paired Student's $t$-test with post-hoc multiple comparisons test) (Fig. 2a).

We analyzed how Prh encodes direction and speed across training. For every training session, neuronal populations ($n = 2335$ neurons, 7 animals) in layer 2/3 (L2/3) of Prh were simultaneously imaged across 2 imaging depths using a multi-area two-photon microscope (Fig. 2b, Supplementary Fig. 6)[31]. In single cells, we observed examples of preferred responses to stimulus direction during early training sessions that disappeared in later sessions (Fig. 2d). We also observed selectivity to stimulus speed emerging over training sessions (Fig. 2e). To characterize these changes at a population level, population decoding was performed on trial conditions related to direction or

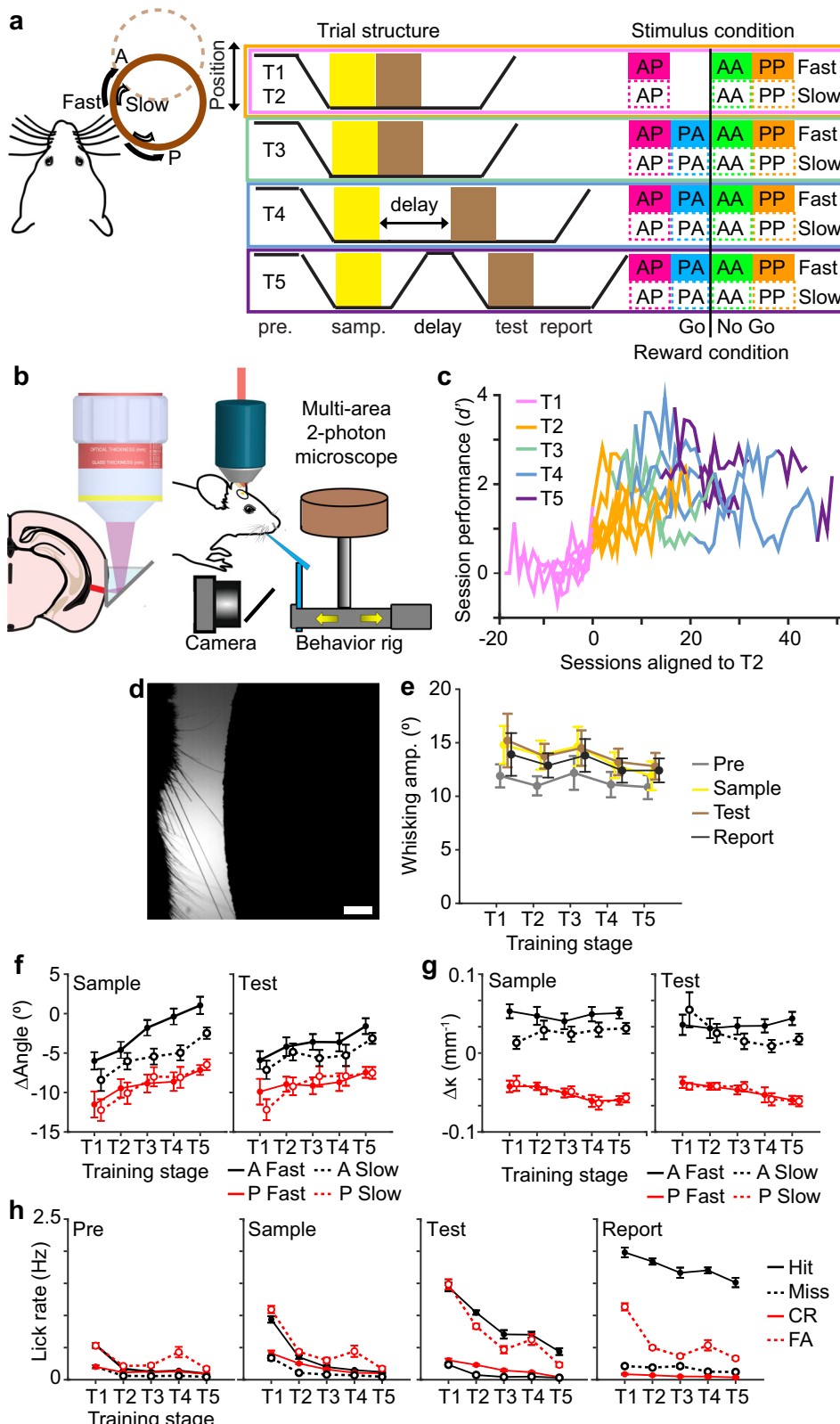

speed (Fig. 2c). Early during learning, direction could be decoded above chance but gradually decreased to chance levels by T5 (Fig. 2f, $P < 1 \times 10^{-8}$, $F_{4,266} = 12.65$, one-way ANOVA with post-hoc multiple comparison test). In contrast, decoders trained to speed increased performance with learning (Fig. 2g, $P < 0.02$, $F_{4,262} = 3.19$, one-way ANOVA with post-hoc multiple comparison test). Overall, this indicates that task training results in weakening representations of task-relevant

stimuli (direction) and strengthening of task-irrelevant stimuli (speed) in Prh.

The above results are in opposition to previous results observed in the primary somatosensory cortex (S1) during task learning which is typified by the strengthening of task-relevant features[30,32]. They are also inconsistent with the changes in whisker kinematics observed across training stages in high-speed videography. We assessed

**Fig. 1 | Measuring behavioral correlates throughout task learning. a** Schematic of an abstract sensory learning task. For home cage task training, animals licked the left port l, or right port (R) for reward for non-match or match stimulus conditions, respectively. For head-fixed task training (2P), non-match stimulus conditions were rewarded (Yes) while match conditions were not (No). During head-fixed task training, animals were primarily trained on directions with fast speeds (95% across T1–T4, 75% for T5) with a smaller fraction of slow speeds trials provided as unexpected stimuli (5% across T1–T4, 25% for T5). **b** Schematic of two-photon imaging of Prh using chronically implanted microprisms allowed during head-fixed task training. **c** Learning curves for individual head-fixed animals trained during two-photon imaging. Only imaged animals reaching T5 were analyzed. **d** High-speed videography was used to measure whisker kinematics during task behavior. **e** Whisking amplitude during each trial period across training stages. **f**, **g** Change in whisker angle (**f**) and curvature (**g**) during sample and test stimulus periods across training stages sorted by speed and direction. **h** Licking rate during each trial period across training stages sorted by choice. Scale bar = 2 mm. Error bars = SEM. $n$ = 90 sessions from 7 animals for (**e**–**h**).

## Table 1 | Home-cage task training stages

| | Performance criteria | NM/M | PA? | Fast/slow | Delay (ms) | Withdraw (cm) |
|---|---|---|---|---|---|---|
| T1 | $d' > 0.45$, 2 sessions | 0.5/0.5 | No | 1/0 | 100 ms | 0 |
| T2 | $d' > 1.68$, 2 sessions | 0.5/0.5 | No | 1/0 | 100 ms | 0 |
| T3 | $d' > 1.68$, 2 sessions | 0.5/0.5 | Yes | 1/0 | 100 ms | 0 |
| T4 | $d' > 1.68/2.05$ (skip) | 0.5/0.5 | Yes | 1/0 | 100–2000 (100 inc.) | 0 |
| | $d' > 1.68/2.05$ (skip) | 0.5/0.5 | Yes | 1/0 | 200–2000 (200 inc.) | 0 |
| | $d' > 1.68/2.05$ (skip) | 0.5/0.5 | Yes | 1/0 | 300–2000 (300 inc.) | 0 |
| | $d' > 1.68/2.05$ (skip) | 0.5/0.5 | Yes | 1/0 | 400–2000 (400 inc.) | 0.1–1.5 (0.1 inc.) |
| | $d' > 1.68/2.05$ (skip) | 0.5/0.5 | Yes | 1/0 | 500–2000 (500 inc.) | 0.2–1.5 (0.2 inc.) |
| | $d' > 1.68/2.05$ (skip) | 0.5/0.5 | Yes | 1/0 | 1000–2000 (500 inc.) | 0.3–1.5 (0.3 inc.) |
| | $d' > 1.68/2.05$ (skip) | 0.5/0.5 | Yes | 1/0 | 1500–2000 (500 inc.) | 0.6–1.5 (0.3 inc.) |
| | $d' > 1.68/2.05$ (skip) | 0.5/0.5 | Yes | 1/0 | 2000 | 0.9–1.5 (0.3 inc.) |
| | $d' > 1.68$ | 0.5/0.5 | Yes | 1/0 | 2000 | 1.2–1.5 (0.3 inc.) |
| T5 | | 0.5/0.5 | Yes | 1/0 | 2000 | 1.5 |

Summary of task settings utilized at each training stage. Performance criteria indicate the behavioral performance necessary to graduate to the next training stages. NM/M indicates the proportion of stimulus conditions belonging to each category. PA indicates whether that stimulus condition was included in the stimulus set. Fast/Slow indicates the proportion of speed stimulus conditions. Delay indicates the starting and ending delay period length along with the interval in which the delay was increased. Withdraw indicates the distance in which the rotor was withdrawn during the delay period, along with the increments of increase.

## Table 2 | Head-fixed task training stages

| | Performance criteria | NM/M | PA | Fast/slow | Delay (ms) | Withdraw (cm) |
|---|---|---|---|---|---|---|
| T1 | $d' > 0.45$, 2 sessions | 0.9/0.1 to 0.5/0.5 over 5 sessions | No | 0.95/0.05 | 100 | 0 |
| T2 | $d' > 1.68$, 2 sessions | 0.5/0.5 | No | 0.95/0.05 | 100 | 0 |
| T3 | $d' > 1.68$, 2 sessions | 0.5/0.5 | Yes | 0.95/0.05 | 100 | 0 |
| T4 | $d' > 1.68$ | 0.5/0.5 | Yes | 0.95/0.05 | 100–2000 (100 inc.) | 0 |
| | $d' > 1.68$ | 0.5/0.5 | Yes | 0.95/0.05 | 200–2000 (200 inc.) | 0 |
| | $d' > 1.68$ | 0.5/0.5 | Yes | 0.95/0.05 | 300–2000 (300 inc.) | 0 |
| | $d' > 1.68$ | 0.5/0.5 | Yes | 0.95/0.05 | 400–2000 (400 inc.) | 0 |
| | $d' > 1.68$ | 0.5/0.5 | Yes | 0.95/0.05 | 500–2000 (500 inc.) | 0 |
| | $d' > 1.68$ | 0.5/0.5 | Yes | 0.95/0.05 | 1000–2000 (500 inc.) | 0 |
| | $d' > 1.68$ | 0.5/0.5 | Yes | 0.95/0.05 | 1500–2000 (500 inc.) | 0 |
| | $d' > 1.68$ | 0.5/0.5 | Yes | 0.95/0.05 | 2000 | 1.5 |
| T5 | | 0.5/0.5 | Yes | 0.75/0.25 | 2000/3000/4000 (0.5/0.25/0.25) prob. | 1.5 |

Summary of task settings utilized at each training stage. Performance criteria indicate the behavioral performance necessary to graduate to the next training stages. NM/M indicates the proportion of stimulus conditions belonging to each category. PA indicates whether that stimulus condition was included in the stimulus set. Fast/Slow indicates the proportion of speed stimulus conditions. Delay indicates the starting and ending delay period length along with the interval in which the delay was increased. Withdraw indicates the distance in which the rotor was withdrawn during the delay period.

whether activity related to direction or speed differed depending on the animals' choice. Decoders were trained on direction or speed separately for correct ('hit' or 'correct rejection') or error ('miss' or 'false alarm') trials. For direction, we found that decoder accuracy during the sample period decreased to chance over learning on correct trials, but this information remained above chance on error trials (Fig. 2h). In contrast, analysis of previously acquired S1 population data in expert animals performing the task showed that direction was stronger on correct compared to error trials (Supplementary Fig. 7). In Prh, decoder performance for speed increased similarly for correct and error trials (Fig. 2i). To more closely examine how speed selectivity relates to choice selectivity in single neurons, we identified neurons with significant population decoder weights to speed (Fig. 2j). We then compared the firing rates of these neurons when sorted for speed conditions versus correct choice conditions. We found examples of neurons that were tuned to both speed and choice (Fig. 2k). We

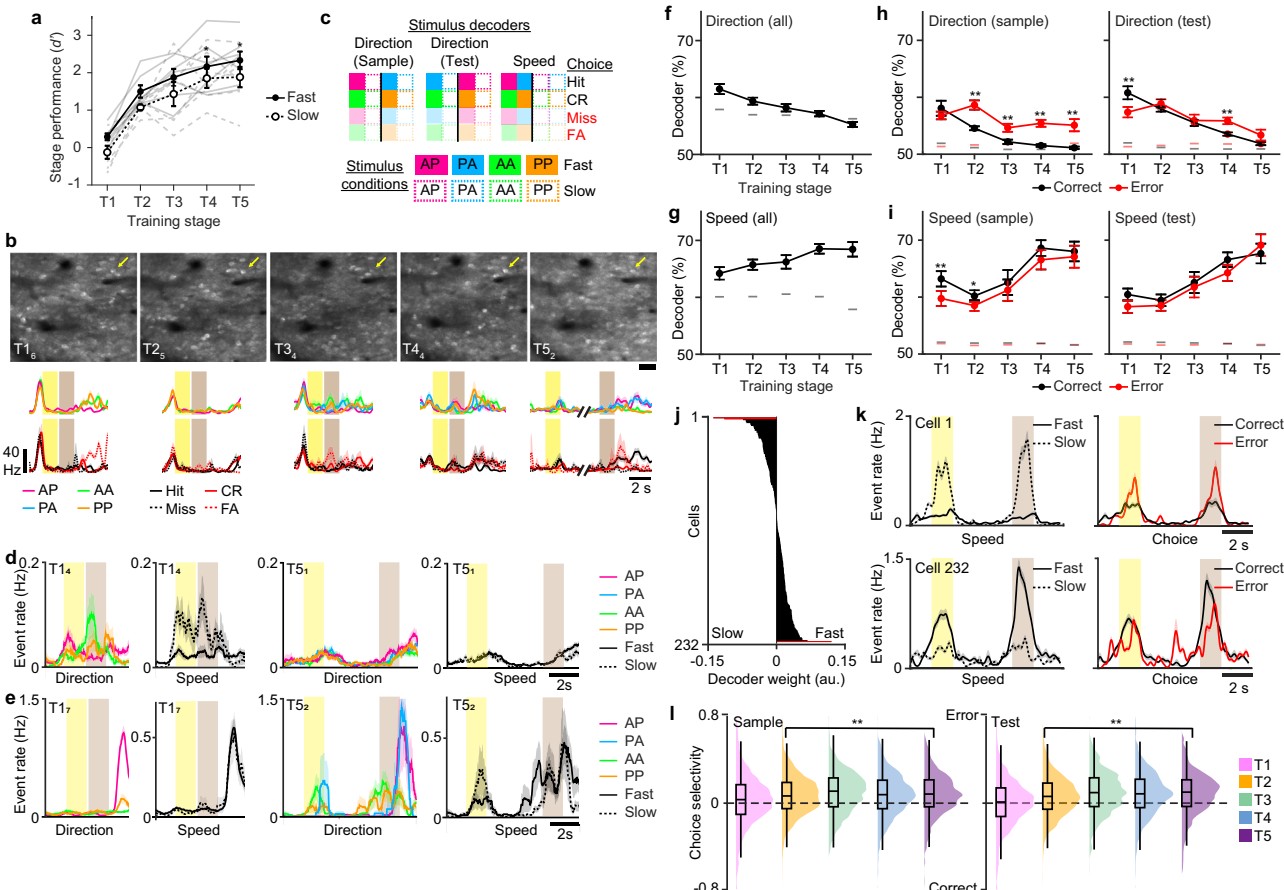

**Fig. 2 | Perirhinal cortex learns sensory prediction errors. a** Behavioral performance across training stages separated by fast versus slow speed trials (two-sided paired Student's *t*-test with post-hoc multiple comparisons test). **b** Example imaging area at denoted training stage and session number, representing $n = 7$ animals imaged for 26–68 consecutive sessions, depending on performance (top row). Mean activity sorted by stimulus condition or choice (bottom row) for indicated neuron (yellow arrow). **c** Schematic of population decoders to stimulus direction or speed. The black line separates decoder trial types. For correct trials, only hit and correct rejection (CR) trials were used. For error trials, only miss and false alarm (FA) trials were used. **d** Example neuron with selectivity to direction and speed during early training sessions ($T1_4$) that shows reduced selectivity in expert sessions ($T5_1$). **e**, Example neuron with developing selectivity to speed in expert sessions ($T5_2$). **f** Decoder performance to stimulus direction across training stages ($P < 1 \times 10^{-8}$, one-way ANOVA with post hoc multiple comparison test). **g** Decoder performance to stimulus speed across training stages ($P < 0.02$, one-way ANOVA

with post-hoc multiple comparison test). **h, i** Decoder performance to stimulus direction (**h**) or speed (**i**) across training stages during the sample (left) and test (right) stimulus period separated by correct versus error trials (two-sided Student's *t*-test). **j** Example population vector weights for the decoder to stimulus speed from one imaging session. Significant weights are indicated (red). **k** Mean event rates, for example, neurons with significant weights in (**j**) sorted by fast versus slow speed trials (left) or correct versus error trials (right). **l** Distribution and box plot depicting mean, s.d., 5th and 95th percentile of choice selectivity during sample (left) or test (right) stimulus period for speed-tuned neurons across training stages (sample period: $P < 1 \times 10^{-15}$; test period: $P < 1 \times 10^{-41}$, one-way ANOVA with post hoc multiple comparison test). Scale bar = 60 µm. Lines indicate the 95th percentile of shuffled performance in (**f**–**i**). Shaded regions = SEM. Error bars = SEM (**f**–**i**). *$P < 0.05$ for (**a**). **$P < 0.005$ for (**f**–**i**). $n = 70$ T1 sessions, 75 T2 sessions, 30 T3 sessions, 79 T4 sessions, 48 T5 sessions from 7 animals for (**f**–**i**). $n = 529$ neurons from 7 animals for (**l**).

measured the choice-selective response distribution of speed-tuned neurons across learning. While the distribution of speed-tuned neurons showed balanced responses to choice during T1, choice selectivity became biased towards error trials once animals demonstrated learned performance (T2–T5) (Fig. 2l, sample: $P < 1 \times 10^{-15}$, $F_{4,7578} = 19.69$, one-way ANOVA with post-hoc multiple comparison tests; test: $P < 1 \times 10^{-41}$, $F_{4,7682} = 50.69$, one-way ANOVA with post-hoc multiple comparison test).

These neural signatures can possibly be explained by Prh's role in familiarity and novel object recognition[33]. Familiarity can be detected by comparing, through subtraction, the current sensory input to one that was previously stored in memory[34]. As sensory information is stored in memory, subtraction results in reduced responses to familiar stimuli and increased responses to novel stimuli. A similar mechanism could be employed for encoding direction and speed during task learning. Memories of direction, as a task-relevant stimulus, may be preferentially stored instead of speed in connected brain areas such

that only that component will be subtracted from the current stimulus when compared in Prh. To illustrate this, we constructed a simple model, focusing on encoding the stimulus features while neglecting models involving working memory[35] or comparison of match and non-match[36], which have been explained previously. The model consists of an autoencoder with input, hidden, and output layers analogous to S1, the hippocampus, and Prh, respectively. The input to the model consists of two stimulus dimensions corresponding to direction and speed (Fig. 3a, left). The network was trained to reconstruct the input in the output layer. An additional output neuron was trained to generate the correct response required to get a reward by reading out from the hidden layer. This additional neuron biased the representation of the hidden layer of the autoencoder to make the direction of motion more relevant than speed. We also limited the activity in the hidden layer by imposing a sparseness constraint (L1-norm) (Fig. 3a, right; see Methods). Finally, analogously to the analysis of Prh activity, a linear classifier reads out the familiarity signal, that is, the difference between the

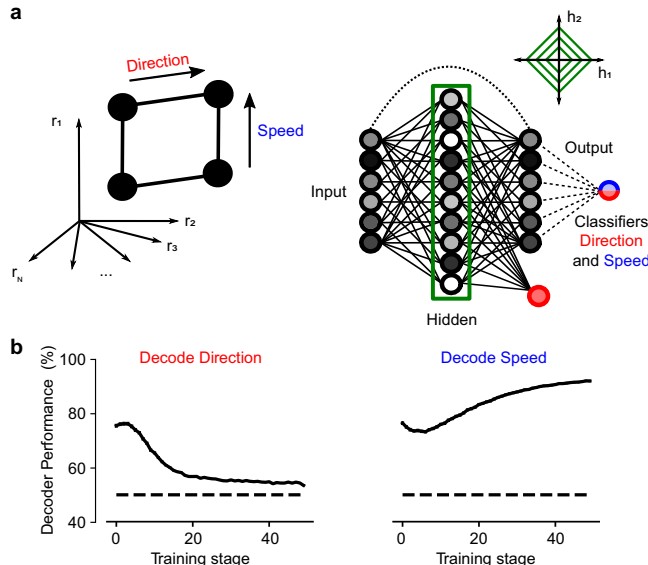

**Fig. 3 | Computational model of sensory prediction errors in perirhinal cortex. a** An autoencoder with three layers (input, hidden, and output) was trained to represent the input. The input consisted of two independent stimulus variables: direction of motion (red) and speed (blue). A linear classifier was trained to decode direction (red) and speed (blue) by reading out the difference between the reconstructed output and the input (dotted line). Sparsity in the hidden layer was imposed by adding an L1-norm term on the loss function. **b** Decoding performance of direction (red, left) and speed (right, blue) as a function of training epoch for the linear classifier reading out from familiarity activity. Similar to experimental results in Prh, decoding performance of direction decreases, whereas decoding performance for speed increases throughout training. Error bars correspond to SEM across independent simulations ($n = 100$). See also Supplementary Fig. 8.

reconstructed output and input[34]. With these simple components, we were able to reproduce the key experimental results observed in Prh. Information about the task-relevant variable direction of motion decreased, whereas information about speed increased throughout learning (Fig. 3b). Importantly, this result was only possible when all components were included in the model (see Supplementary Fig. 8).

Overall, the results above indicate that Prh does not represent sensory information in the same manner as S1 does. Instead, it suggests that stimulus activity in Prh may reflect a sensory prediction error signal (i.e., the difference between actual and expected sensory information), consistent with theories of predictive coding[3] and Prh's role in familiarity and novel object recognition. Information about direction decreases as Prh forms an internal model of direction as the task-relevant feature, explaining away the delivered stimuli. Concurrently, information about speed increases to signal the prediction error between directions that are presented at the expected fast speeds versus the unexpected, weak, slow speeds.

**Stimulus-reward associations emerge and stabilize with learning**
To understand how sensory and reward information are integrated to form stimulus-reward associations, we analyzed how representations of reward outcomes evolved across learning. A cross-session decoder was trained using Hit vs. non-Hit trials from one session and tested on other sessions across learning (Fig. 4a). When assessing cross-session performance between neighboring sessions during the reporting period, representations of reward outcome were stably represented above chance on a session-to-session basis. No differences in session-to-session performance were found between training stages (Fig. 4b, $P = 0.19$, $F_{4,260} = 1.54$, one-way ANOVA). Analysis of cross-session performance across longer time scales and across training stages showed that representations of reward outcome were less stable early in

training (T1) but stabilized as animals learned the task (Fig. 4c). These results suggest that learning produces a stable, long-term representation of reward outcome.

Given that reward outcome stabilizes with learning, we asked whether such representations reflect a stimulus-reward association that would precede reward delivery. A cross-temporal decoder was trained on Hit vs. non-Hit trials during the report period and then tested on time points across the trial period. We identified a gradual retrograde expansion of decoder performance related to reward outcome over the course of learning that preceded reward and extended into the test stimulus period (Fig. 4d). Analysis of the onset of decodable reward outcome across training stages showed that this expansion emerged as animals demonstrated learned performance (T2) and continued to expand throughout the additional training stages (Fig. 4e, $P < 0.002$, $F_{4,282} = 4.44$, one-way ANOVA with post-hoc multiple comparison test). To test whether this temporal expansion is specific to rewarded trials, we conducted a similar analysis of cross-temporal decoders trained to non-rewarded conditions that controlled for either licking behavior (false alarm) or correct choice (correct rejection). Neither decoder showed onset accuracy that extended into the test period. This demonstrates that neural representations on Hit trials correspond to a stimulus-reward association. The temporal profile of this expansion suggests that this association emerges in a retrograde manner from the reward outcome.

**Stimulus-reward associations generalize in an abstract format**
We next asked whether stimulus-reward associations were specific to a given stimulus set or could generalize across stimulus conditions. To address this, we analyzed how representations changed from T2 to T3 when the novel PA stimulus-reward contingency was introduced. Behaviorally, mice were flexibly able to respond correctly on the first session in which PA was introduced ($T3_0$). Performance on PA further improved over ~4–5 sessions, reaching similar levels as AP (Fig. 5a). We observed examples of single cells that exhibited distinct temporal responses between AP and PA conditions at $T3_0$. These responses changed over sessions, resulting in similar responses between the two conditions (Fig. 5b). To characterize these changes at a population level, we trained two separate population decoders on activity during the reporting period on rewarded conditions using either only AP or PA (Fig. 5c). This allowed us to independently evaluate each representation across T3 sessions. Cross-temporal analysis showed that the temporal profile of AP and PA representations were distinct at $T3_0$ but became similar after 4 sessions ($T3_4$) (Fig. 5d). Whereas the onset accuracy extended into the test period for AP at $T3_0$, indicative of a stimulus-reward association, onset accuracy for PA initially was restricted to the reporting period but expanded into the test period over the course of 3–4 sessions (Fig. 5e, $P < 0.002$, $F_{9,54} = 3.64$, two-way repeated measures ANOVA with post-hoc Student's $t$-test). While licking behavior also differed between AP and PA trials at $T3_0$ and later became similar over multiple sessions, neural signals followed licking behavior (Supplementary Fig. 9). The time lag between licking and stimulus-reward association was inconsistent across conditions and sessions. This demonstrates that the acquisition of new stimulus-reward contingencies occurs through a common mechanism of retrograde expansion from reward outcome in a manner that cannot be explained by motor behavior.

Representations of AP-reward and PA-reward associations could exist in different or similar neural subspaces. The latter would imply that the geometry of stimulus-reward associations in Prh is represented in an abstract format[7]. To test this, we analyzed the cross-condition performance for each of the two separate population decoders (ie. testing AP performance using a PA decoder and vice-versa). Cross-condition PA performance to the AP decoder during the test stimulus period was initially worse than the opposite cross-condition but gradually improved over the course of 9 sessions

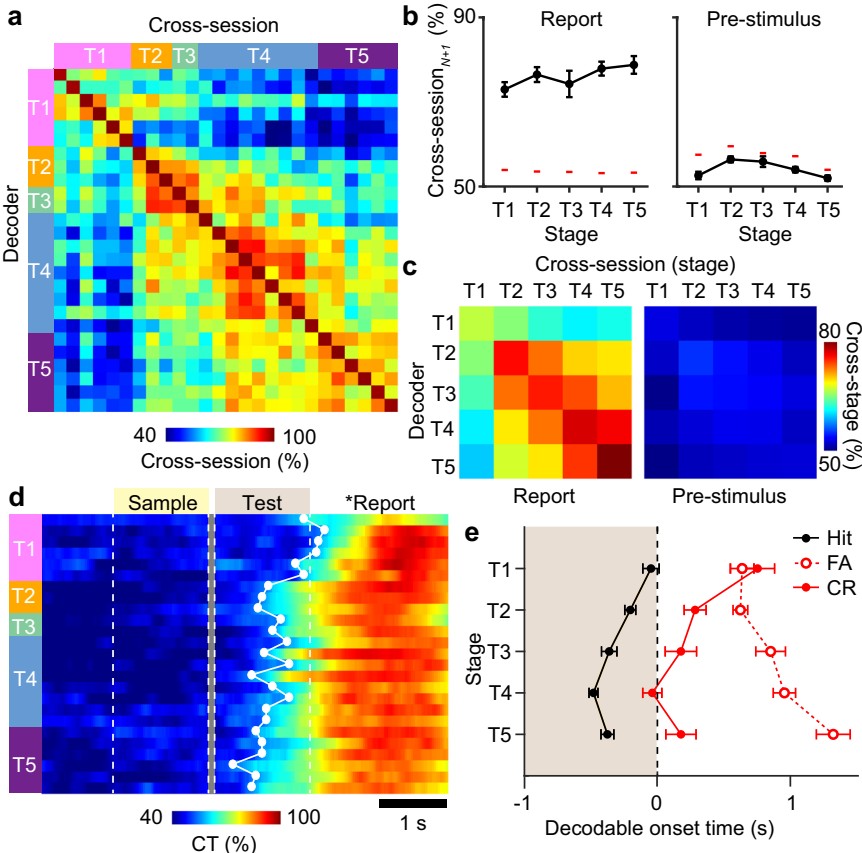

**Fig. 4 | Learning of stimulus-reward associations. a** Example of cross-session Hit vs. non-Hit trial decoder performance trained on activity during the reporting period for the session and tested on all other sessions for one animal across training. **b** Cross-session performance for Hit trial decoders trained on session$_N$ and tested on session$_{N+1}$ for report activity (left) or pre-stimulus activity (right) across training stages. **c** Cross-session decoder performance across training stages for report activity (left) or pre-stimulus activity (right). **d** Example of cross-temporal (CT) decoder for reward conditions trained on report activity across each training session for one animal. The first decodable time point above chance is shown (white dot). **e** Decodable onset timepoint for the cross-temporal decoder of report activity for decoders trained on hit, false alarm, or correct rejection trials ($P < 0.002$, $F_{4,282} = 4.44$, one-way ANOVA with post-hoc multiple comparison test. Error bars = SEM. Red lines indicate the 95th percentile of shuffled performance (**b**). $n = 70$ T1 sessions, 75 T2 sessions, 30 T3 sessions, 79 T4 sessions, 48 T5 sessions from 7 animals.

(Fig. 5f, g, $P < 0.05$, $F_{9,54} = 2.16$, two-way repeated measures ANOVA with post hoc Student's $t$-test). This suggests acquisition of new stimulus-reward contingences occurs in two phases: an initial establishment of the stimulus-reward association followed by a consolidation that aligns the new association into the same subspace of existing stimulus-reward associations. Overall, these findings demonstrate that Prh can generalize across novel stimulus-reward contingencies to form stimulus-reward associations that are representationally abstract.

**Neural signatures of expected outcome in Prh**

The observation that stimulus-outcome associations emerge in a retrograde manner to precede the reporting period suggests that stimulus information is integrated with an ongoing activity that could signal an expected outcome (i.e., reward delivery). Neural activity reflecting the expectation of reward or punishment has been observed during task engagement in other brain areas[37]. Therefore, we asked whether ongoing Prh activity during the trial period could contain an expectation signal of future outcomes. We define the expected outcome as the ability of a linear decoder to decode trial outcomes when trained on the activity at the beginning of the trial (pre-stimulus period). To look for evidence of population activity corresponding to the expected outcome, two separate population decoders were trained on either hit vs. non-hit trials (Expected Hit) or correct rejection (CR) vs. non-CR trials (Expected CR) during the pre-stimulus period (Fig. 6a). When trained and tested during the pre-stimulus period (Fig. 6b), we surprisingly found that trial outcome could be decoded above chance throughout training. Expected Hit could also be decoded when trained on activity during the sample and test period (Supplementary Fig. 10). The accuracy of these decoders trained during the pre-stimulus, sample, and test period was consistently weaker (~60-70%) compared to decoders trained during the reporting period (~80-90%) (Fig. 6d). Decoders applied to different trial periods could reflect distinct population subspaces with their own properties. While cross-session decoders trained during the reporting period stabilized with training, indicative of forming a long-term stimulus-reward association, cross-session decoders during the pre-stimulus period were unstable and not able to perform above chance (Fig. 4b, c). This suggests that the expected outcome activity reflects something other than a stably encoded representation.

Rather than encoding an experienced representation, we hypothesized that the expected outcome could reflect an anticipatory network state. If persistent across the length of a trial, it could serve to link task events to a given outcome. We assessed the persistence of network activity as the ability of a decoder trained during the pre-stimulus period to cross-temporally decode trial outcomes when tested on activity during the reporting period. Interestingly, we found performance on experienced outcome activity to be below chance levels (ie. below the 5th percentile of the shuffled distribution) (Fig. 6c). This was particularly strong during T2-T5 sessions when animals exhibited strong task performance (Expected Hit: $P < 1 \times 10^{-5}$,

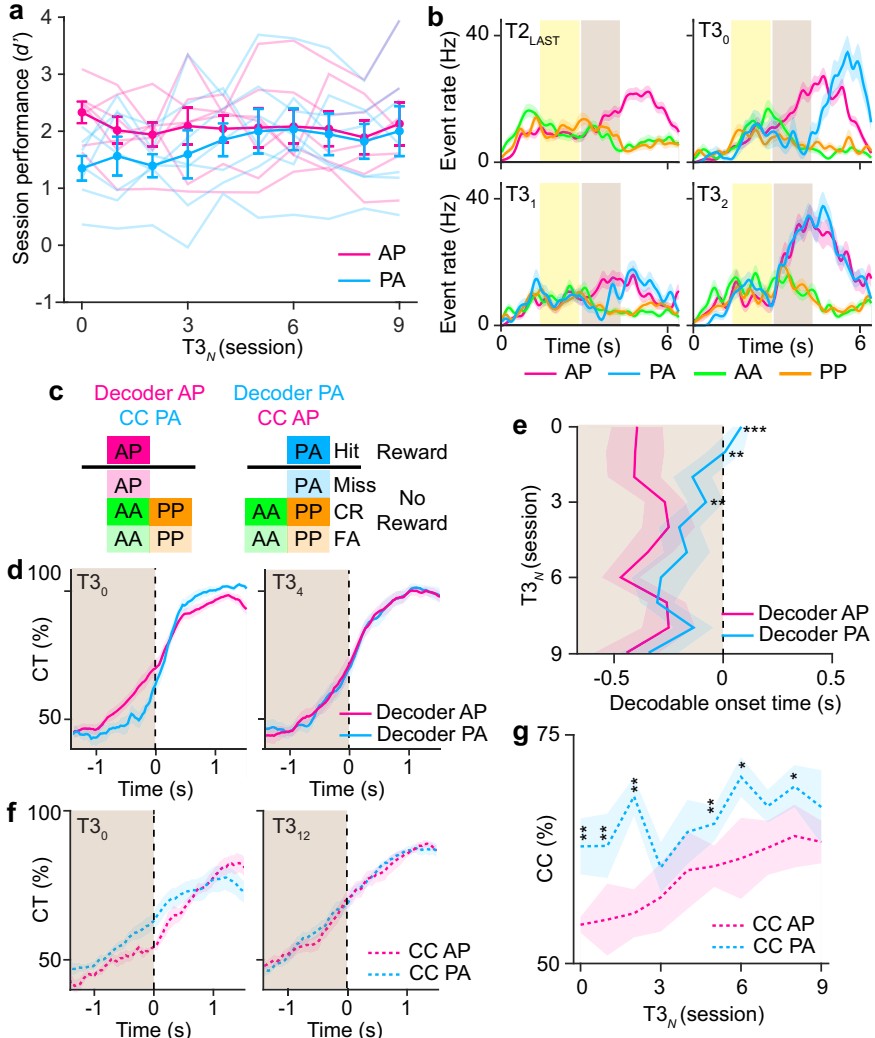

**Fig. 5 | Stimulus-reward associations are abstract. a** Behavioral performance aligned to the first T3 session for AP versus PA stimulus conditions. Mean and individual animal performance is shown. **b** Mean activity in an example neuron separated by stimulus conditions across the first four T3 sessions. **c** Schematic for population decoder for reward using either only AP or PA stimulus conditions. Cross-condition (CC) decoder is also shown for the complementary condition. **d** Mean cross-temporal (CT) decoder performance trained on report activity for the rewarded AP or PA condition during the $T3_0$ or $T3_4$ session. **e** Mean decodable onset timepoint for either the rewarded AP or PA condition T3 sessions ($P < 0.002$, two-way repeated measures ANOVA with post-hoc Student's $t$-test). **f** Mean cross-temporal decoder performance trained on report activity for the rewarded AP or PA condition and tested on the cross condition during the $T3_0$ or $T3_{12}$ session. **g** Mean cross-temporal decoder performance trained on report activity for the rewarded AP or PA condition and tested on the cross-condition test period activity across T3 sessions ($P < 0.05$). Error bars = SEM. Shaded regions = SEM. *$P < 0.05$, **$P < 0.02$, ***$P < 0.001$ for (**e**) and (**g**). $n = 7$ animals for (**a**, **d–g**).

$F_{4,296} = 7.64$; Expected CR: $P < 1 \times 10^{-19}$, $F_{4,285} = 29.18$, one-way ANOVA with post-hoc multiple comparisons test).

To better understand how pre-stimulus activity predicts outcome activity below chance in single neurons, we identified neurons with significant population decoder weights. These neurons exhibited low levels of activity during the pre-stimulus period that differed slightly when sorted between Hit, Miss, FA, and CR trials. One neuron that showed slightly elevated pre-stimulus activity on CR trials showed robust outcome responses on Hit trials. Another neuron that showed slightly elevated pre-stimulus activity on Hit trials showed robust outcome responses on CR trials (Fig. 6h). We examined the population trajectory along the subspace of the pre-stimulus decoder (Fig. 6i). For Expected Hit, the population activity was projected along the decision variable axis for each of the 4 choice conditions over the time course of the trial. We observed that activity on hit and non-hit trials was separated along the axis through the pre-stimulus and sample stimulus period. The trajectories converged during the test stimulus period and then flipped their sign

during the reporting period. This suggests that the decoder trained on expected outcomes captures neurons whose firing initially favors one potential trial outcome during the pre-stimulus period but later reverses its response to prefer the actual outcome during the reporting period. The sign flip along this subspace explains the below chance performance during the reporting period.

To confirm that activity in the pre-stimulus period constitutes a prospective and not a retrospective signal, we analyzed the performance of several cross-temporal decoders. A cross-temporal decoder trained during the reporting period was not able to explain reward information during the pre-stimulus period (Fig. 6e). To test if pre-stimulus information reflects a trial history of recent outcomes as observed in other cortical areas[38], cross-temporal decoders between the pre-stimulus and the reporting period of the previous trial were tested (Fig. 6f, g). These decoders did not perform above chance. Overall, this demonstrates that activity early in the trial constitutes a prospective signal whose subspaces emerge with training to link expectation to learned outcomes.

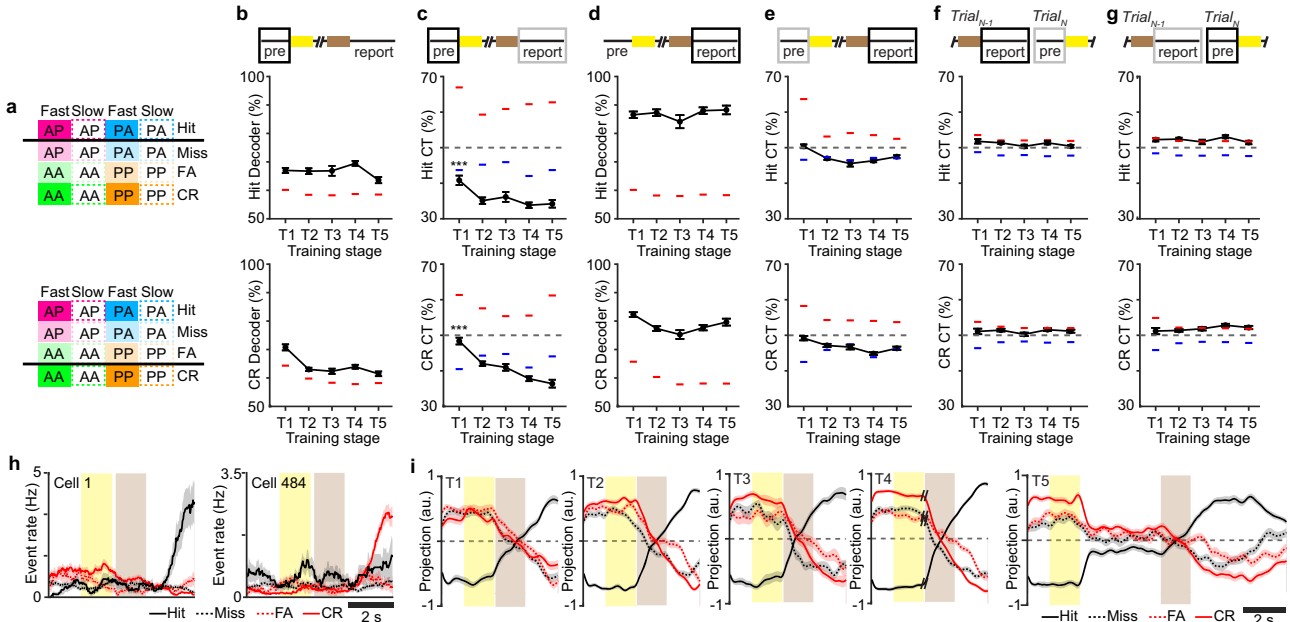

**Fig. 6 | Perirhinal cortex encodes expected outcomes throughout task learning. a** Schematic of the population decoder trained to Expected Hit (top) or Expected CR (bottom). **b** Decoder performance to Expected Hit (top) and Expected CR (bottom). **c** Cross-temporal (CT) decoder performance to Expected Hit (top) and Expected CR (bottom). The black box indicates the trained time window during the pre-stimulus period. The gray box indicates (solid box) the tested time window during the reporting period. **d** Decoder performance during the reporting period for Hit (top) and CR (bottom). **e** CT decoder performance trained during the reporting period (black box) and tested during the pre-stimulus period (gray box) for Hit (top) and CR (bottom) trials. **f** CT decoder performance trained during the reporting period (black box) and tested during the pre-stimulus period of the following trial (gray box) for Hit (top) and CR (bottom) trials. **g** CT decoder performance trained during the pre-stimulus period (black box) and tested during the

reporting period of the previous trial (gray box) for Hit (top) and CR (bottom) trials. **h** Mean estimated firing rate, for example, neurons with significant weights for Expected Hit decoder. Cell 1 shows elevated firing during the pre-stimulus period on CR trials but strongly responds during the reporting period of Hit trials. Cell 484 shows elevated firing during the pre-stimulus period on Hit trials but strongly responds during the report period of CR trials. **i** Mean projection of neural activity along the decision variable for Expected Hit (**c**) across the trial period sorted by trial type across training stages. Shaded regions = SEM. Error bars = SEM, ***$P < 1 \times 10^{-3}$, one-way ANOVA with post-hoc multiple comparisons test. For **b**–**g**, dashed lines indicate the 95th percentile (red) and 5th percentile (blue) of nulled performance for the classifier after shuffling trial labels. $n = 70$ T1 sessions, 75 T2 sessions, 30 T3 sessions, 79 T4 sessions, 48 T5 sessions from 7 animals.

## Cholinergic signaling mediates expected outcome calcium signals

Our observations suggest that expected outcome signals more likely reflect a persistent network state analogous to task engagement rather than some intentional prediction of a future outcome. To identify mechanisms that could drive such activity, we considered Acetylcholine (Ach), a major neuromodulator known to alter neural network excitability[22] and to be associated with task engagement and reward expectation[39]. To visualize Ach activity during the early stages of training (T1 and T2), we imaged Ach release in Prh using the fluorescent Ach indicator GRAB-Ach3.0[40] (Fig. 7a). Bulk Ach signals were measured across the field of view. Prominent high Ach release was measured during the pre-stimulus period across all trials (Fig. 7b, c, Supplementary Fig. 11). On hit trials, increases in Ach was also observed to be related to licking behavior prior to reward delivery but not during reward consumption. Similar dynamics were observed on false alarm trials when no reward was delivered. These dynamics suggest that Ach in Prh signals behavioral correlates of reward expectation. To quantify the relationship between Ach and the behavioral task, we modeled Ach signals using a generalized linear model (GLM) with task variables representing the pre-stimulus period, stimulus direction, pre-reward licking, post-reward licking, reward delivery, and post-trial period (Fig. 7d, e, Supplementary Fig. 12). The pre-stimulus task variable best explained Ach signals and increased from T1 to T2 (Fig. 7f, $P < 0.05$, Student's $t$-test). This increase in pre-stimulus Ach coincided with the emergence of sustained expected outcome signals (Fig. 6c).

Ach can modulate neuronal activity either in a transient or sustained manner via nicotinic (nAch) or muscarinic (mAch) receptors,

respectively[22]. To determine if the sustained expected outcome depends on a specific Ach receptor, two-photon calcium imaging was performed on animals trained up through T2. Using reversible pharmacological treatments, population activity was monitored while nAch or mAch receptors were inactivated using systemic delivery of mecamylamine or scopolamine, respectively. Inactivation occurred in alternating imaging sessions that were additionally interleaved with control recovery sessions (Fig. 7g). We found that systemic administration of scopolamine, but not mecamylamine, significantly impaired task performance (Fig. 7h, $P < 1 \times 10^{-4}$, Student's $t$-test). Population activity was also disrupted. Using a cross-temporal decoder trained on Hit vs. no-hit trials during the reporting period, we find that scopolamine treatment weakened stimulus-reward associations by both lowering decoder performance and delaying the onset of decodable reward outcome ($P < 0.02$, Student's $t$-test) (Fig. 7i, j).

We next examined how nAch or mAch receptor inactivation affected expected outcome activity and how those activity patterns related to task performance during T2. Under control conditions, we observed a correlation between expected outcome decoder performance and behavioral performance on a per-session basis for both Expected Hit and Expected CR (Fig. 7k, l). Cross-temporal decoder performance tested on the report period was also correlated with behavior performance for Expected CR (Fig. 7n) but not for Expected Hit (Fig. 7m). Scopolamine, but not mecamylamine, disrupted the relationship between decoder performance and task performance across all conditions (pre-stimulus Expected Hit: $R = 0.08$, $P = 0.83$, pre-stimulus Expected CR: $R = 0.23$, $P = 0.54$, report Expected CR: $R = 0.16$, $P = 0.65$, Pearson's correlation). Overall, this demonstrates

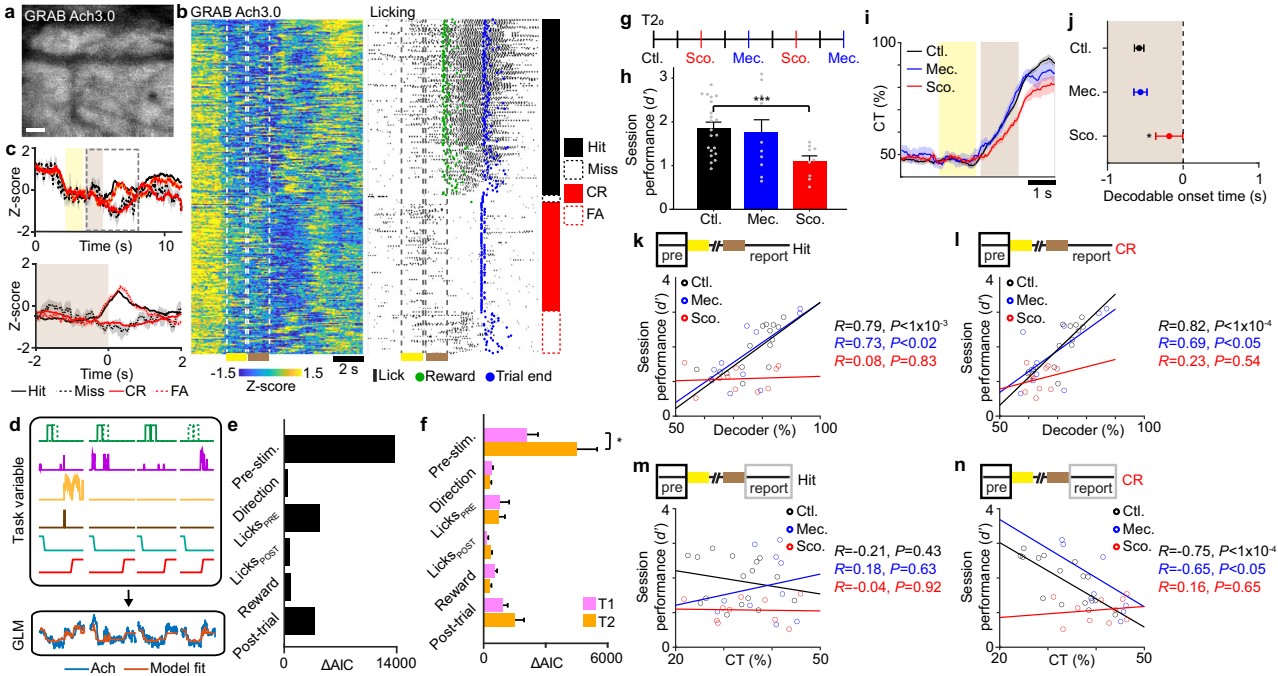

**Fig. 7 | Expected outcome depends on acetylcholine signaling. a** Two-photon images of GRAB-Ach3.0 expression in the perirhinal cortex, representing $n = 4$ animals imaged for 12–42 consecutive sessions, depending on experimental conditions. **b** Example bulk Ach signals (left) and licking behavior (right) sorted by trial type for one session. Sample (yellow) and test (brown) periods are indicated. Timepoint of stimulus delivery, reward, and the end of the trial are also indicated. **c** Mean Ach signals across the trial period separated by the choice-aligned beginning of the trial (top). The bottom panel shows a magnified view of signals (dashed rectangle in the top panel) aligned to the behavioral report. **d** Schematic of GLM depicting basis functions for task variables (top) applied to model Ach signals (bottom). **e** Example encoding of task factors from imaging session shown in (**b**). **f** Encoding of task factors across T1 and T2 sessions. **g** Schematic of T2 calcium imaging sessions alternating between control no inactivation (Ctl), nAch receptor inactivation by mecamylamine (Mec.), and mAch receptor inactivation by

scopolamine (Sco.). **h** Task performance across pharmacological inactivation sessions. **i** Stimulus-reward association determined by cross-temporal (CT) decoder performance for Hit vs. non-Hit trials across pharmacological conditions. **j** Decodable onset timepoint for Stimulus-reward association for (**i**) across pharmacological conditions. **k** Correlation between task performance and decoder performance to Expected Hit across pharmacological conditions. **l** Correlation between task performance and decoder performance to Expected CR across pharmacological conditions. **m** Correlation between task performance and cross-temporal (CT) performance to Expected Hit across pharmacological conditions. **n** Correlation between task performance and cross-temporal (CT) performance to Expected CR across pharmacological conditions. Shaded regions = SEM. Error bars = SEM. Scale bar = 20 µm. *$P < 0.05$, **$P < 0.01$, ***$P < 1 \times 10^{-4}$. $n = 29$ T1, 26 T2 sessions from 4 animals for (**f**); $n = 4$ animals, 19 Ctl., 10 Mec., 11 Sco. sessions for (**g–n**).

that mAch receptor-mediated signaling is involved in establishing sustained expected outcome activity in Prh and that expected outcome activity is necessary for correct task performance.

## Discussion

In summary, we demonstrate how Prh is involved in learning an internal model of sensory-guided task behavior that we refer to as a predictive map. Our analysis of sensorimotor variables during head-fixed conditions, along with Prh activity, as described below, indicates that Prh neurons do not encode sensory and motor information in a direct, bottom-up manner as observed in primary somatosensory cortex[28,30,32]. Instead, we propose that sensory information is transformed in Prh into a predictive map that is reflected in three forms of activity: (1) sensory prediction errors; (2) stimulus-outcome associations, and; (3) expected outcome signals (Supplementary Fig. 13).

Sensory prediction errors reflect the learning of task-relevant stimulus features. We show that information about stimulus direction—a task-relevant feature—decreases with learning but is still present in error trials. Stimulus speed information—corresponding to the strength in stimulus direction—increases with learning and is accompanied by higher firing rates on error trials. These changes in learning are consistent with theories of predictive coding in which neurons signal the difference between expected and actual sensory information[1]. We speculate that Prh evaluates an internal model of task-relevant stimuli via the hippocampus against ongoing stimuli

information from sensory neocortex resulting in signals that reflect sensory prediction errors. These results are consistent with previous studies attributing Prh's role in novel object recognition memory[20,21], wherein familiarity is learned from repeated exposure to objects such that novel objects signal the prediction error between experienced and familiar stimuli. In our experimental design, animals experienced slow directions at lower frequencies than fast directions. This does not allow us to disambiguate whether the sensory prediction error signals we observe are driven by familiarity due to stimulus probability or task-dependent feature learning. We used a limited computational model developed for familiarity detection[34] to specifically test our sensory prediction error results. The ability of the model to recapitulate those results suggests that sensory prediction errors due to familiarity or task learning could arise from similar mechanisms.

Sensory prediction errors in Prh may serve two purposes. First, they may act as a teaching signal that promotes updating of task-related variables through error-driven learning that functions to minimize differences between actual and expected sensory information[11]. This would produce a more accurate internal model of task-relevant sensory features. Second, considering feedback connections from Prh back to the sensory neocortex, prediction errors may aid in sensory inference by boosting bottom-up sensory information in lower areas under circumstances of discrepant sensory signals to help guide behavior[41]. Our results suggest a relationship

between the strength of prediction error signals and incorrect choice behavior. Inference may help to support feature-invariant encoding of task-relevant stimuli (ie. encoding direction invariant to speed).

While stimulus features that are necessary but not sufficient to predict outcome are encoded as sensory prediction errors, combined features that are sufficient to predict reward are encoded as stimulus-reward associations. Through task learning, stimulus-reward associations stabilize and expand in a retrograde manner from the time of reward back to the test period. These signals show similarity to goal-approach neurons in the medial entorhinal cortex and hippocampus during spatial navigation behavior, which increases their activity as animals approach learned locations of reward[42]. This representation generalizes to novel stimulus-reward contingencies. New associations distinctly emerge through a similar mechanism of retrograde expansion. The novel contingency then geometrically aligns with existing associations into an abstract format[7]. This demonstrates that predictive maps can flexibly adapt to newly encountered stimulus-reward contingencies.

Finally, we observe sustained network activity that links prospective signals of expected outcomes with the experienced outcome. We speculate that expected outcome signals facilitate learning and recall of sensory-related task models[20,21,43]. These signals, along with stimulus-reward associations, depend on cholinergic signaling. Through GRAB-Ach3.0 imaging, we find that Ach is transiently released at the beginning of each trial to establish a task-specific expected outcome state space. This transient release could have a sustained impact on neuronal excitability and persistent network activity over the course of the trial via mAch receptor activation[22]. We find that systemic blockade of both nAch and mAch receptors disrupts expected outcome activity while blockade of mAch receptors disrupts task performance. High cholinergic tone has been associated with an encoding-like "external" mode of processing in the hippocampus and neocortex, while low Ach is associated with a retrieval-like "internal" mode of processing[24]. We propose that Ach-associated, expected outcome activity may enable sensory information to be evaluated against internal models underlying prediction coding and error-driven learning, consistent with an external mode of processing. Once sensory evidence is sufficient to predict reward, the network switches to a retrieval-like "internal" mode in which stimulus-reward associations are retrieved from long-term memories ascribed to cognitive maps. Given the timescale of these network switches, additional neuromodulators with the capacity to alter network excitability, such as serotonin, norepinenphrine, or dopamine, may also be involved and worth further investigation. Overall, a predictive map of task behavior could emerge from these switches in network states that engage other brain areas and allow error-driven and associative plasticity to guide model learning in local circuits.

## Methods

### Mice
Experiments in this study were approved by the Institutional Animal Care and Use Committee at Boston University and conform to NIH guidelines. Head-fixed behavior experiments were performed using male C57BL/6 mice (Charles River Laboratories). Home cage behavior experiments used a male ($n = 16$) and female ($n = 6$) C57BL/6 mice (Charles River Laboratories). All animals were 6–8 weeks of age at the time of surgery. Mice used for behavior were housed individually in reverse 12-h light cycle conditions with standard ranges for temperature (68–79°) and humidity (30–70%). All handling and behavior occurred under simulated nighttime conditions.

### Animal preparation
For surgical procedures, animals were anesthetized with 1–3% isoflurane. Prh was targeted stereotaxically (2.7 mm posterior to bregma,

4.2 mm lateral, and 3.8 mm ventral). For inactivation experiments, bilateral injections were targeted via the parietal bone. For each side, animals received either retroAAV-*hSyn-Cre* ($4.5 \times 10^{12}$ vg/mL) and AAV9-*hSyn-dio-hM4Di-mCherry* ($6.0 \times 10^{12}$ vg/mL) (1:1, 600 nL) or retroAAV-*hSyn-Cre* and AAV9-*hSyn-dio-mCherry* ($6.0 \times 10^{12}$ vg/mL) (1:1, 600 nL). For tracking in the home cage training, a radio frequency identification (RFID) glass capsule (SEN-09416, Sparkfun) was implanted subcutaneously in the animal's back. For in vivo imaging experiments, a unilateral injection was targeted via the temporal bone at 250 μm and 500 μm below the pial surface of either AAV.PHP.eB-*EF1α-RCaMP1.07* (600 nL, $6 \times 10^{12}$ vg/mL), AAV9-*hSyn-GRAB-Ach3.0* (600 nL, $2.5 \times 10^{12}$ vg/mL), or AAV2-retro-*CAG-GFP* (600 nL, $1 \times 10^{12}$ vg/mL). For optical access, an assembly consisting of a 2 mm aluminum-coated microprism (MPCH-2.0, Tower Optical) adhered to coverglass along the hypotenuse, and the side facing Prh was implanted over the pial surface. A metal headpost was implanted on the parietal bone of the skull to allow for head fixation. For unilateral retrograde tracing between Prh and S2, CTB-Alexa647 (Molecular Probes, Invitrogen; 300 nL, 1% wt/vol) was delivered into Prh, targeted via the temporal bone and CTB-Alexa488 (300 nL, 1% wt/vol) was delivered into S2 (0.7 mm posterior to bregma, 4.2 mm lateral, 250 and 500 μm below the pial surface).

### Home cage task training
Two weeks after injections, animals were trained in a whisker-based context-dependent sensory task adapted for training in an automated live-in environment (Supplementary Note S1). The animals were singly housed in individual cages. Three cages were attached to a shared training system wherein individual access was restricted via servo-operated doors (SG92R, Tower Pro) controlled by a microcontroller (Uno Rev3, Arduino). The training system consists of a narrow corridor that restricts body and head movement at the front of the corridor where the sensory stimulus is delivered. Equipment for whisker stimulus, lick detection, sound delivery, air puff delivery, and water delivery was similar to as described[28]. Water ports were attached to a capacitive lick sensor (AT42QT1010; SparkFun) that dispenses 5–6 μL of water through a miniature solenoid valve (LHDA0531115H; The Lee Company). For the rotation stimulus, commercial grade sandpaper (3 M; roughness: P100) was mounted along the outside edge of a 6 cm diameter rotor, attached to a stepper motor (Zaber) to deflect the whiskers, which was mounted onto a linear stage (Zaber) to place the rotor within whisker reach. Two lick ports were mounted onto a linear actuator (L12-P, Actuonix) that controlled access to water during the task. An LED beam breaker (2167, Adafruit) at the head of the training system such that animals self-initiated behavioral trials by breaking the beam with their body.

Each animal was provided access to the training system via the servo door through scheduled two-hour morning and two-hour afternoon session blocks. Animals were initially acclimated by learning to retrieve water from the lick ports. Once acclimated, animals proceeded to task training. During task training, the rotor providing the whisker stimulus was retracted during the inter-trial interval and placed in reach during stimulus periods. The lick spouts were only presented during the reporting period and retracted at all other times. A two-forced alternative choice task design was used in which the correct choice required licking to the right port for non-match stimuli and to the left port for match stimuli. Only fast rotations (1.75 cm/s) of stimulus direction were used.

The training was divided into 5 stages (T1–T5) (Table 1, Supplementary Note S3). For T1 and T2, one non-match stimuli (AP) and two match stimuli (AA, PP) were included. T1 was defined as initial naïve performance. T2 was defined as learned performance beginning from the point in which animals displayed $d' > 0.45$ for two consecutive sessions. For T3, the second non-match stimuli (PA) was introduced. For T4, delays between the sample and test stimuli were gradually

 

lengthened up to 2 s. The rotor was also gradually retracted up to 1.5 cm out of whisker reach. T5 was defined as consistent expert performance with a 2 s delay and 1.5 cm rotor retraction. Advancement from T2-T5 was automated based on behavioral performance of two consecutive sessions of >80% correct ($d'$ ~ 1.68). The delay period and rotor withdrawal distance during T4 were automatically increased based on behavioral performance of >80% correct ($d'$ ~ 1.68) across a 15-trial sliding window.

In addition to the water reward, the correct behavioral choice was reinforced using three automatically adjusted task settings (Table 3, Supplementary Note S4). Punishment in the form of a combination of time outs (2–10 s) and air puffs to the face was introduced to discourage incorrect decisions. Time outs ranged from 2 to 10 s. Air puffs (100 ms) ranged from 1 to 5 trains and were introduced for >7 s time out. Punishment systematically increased during the poor performance, corresponding to <70% correct ($d'$ ~ 1.05) over a 50-trial sliding window. Punishment was automatically decreased if the proportion of misses in this window exceeded 50%. To correct for report biases in which the animal repetitively licked one port irrespective of stimulus condition, the probability of match vs. non-match stimulus conditions was increased in favor of the stimulus condition associated with the neglected spout. To correct for primacy and recency stimulus bias resulting in disproportionally greater error trials for one of the two match conditions or one of the two non-match conditions, the probability of one of the two match or non-match conditions was adjusted in favor of the condition with the greater proportion of errors.

For chemogenetic inactivation, Compound 21 (HB6124, HelloBio) was provided in the drinking water (9.5 μg/mL H$_2$O, 1 mg/kg body weight). Animals only received water by performing the task. Their weight was monitored daily to ensure body weight did not drop below 80% of initial weight. Animals were trained continuously for 6 weeks.

## Head-fixed task training
Two weeks after microprism implantation and injections, animals were handled and acclimated to head fixation. Training to a head-fixed whisker-based context-dependent sensory task was performed similar to as described[28] (Supplementary Note S2). Water ports and stimulus delivery hardware were the same as the home-cage training system. Whiskers were trimmed to a single row for videography. Animals are trained for two sessions per day. A go/no-go task design was used in which animals licked for water reward for non-match stimulus

conditions and withheld licking for match stimulus conditions. T1-T3 training stages were similar to stages defined in home cage task training (Table 2). For T4, the delay between sample and test stimuli was gradually increased from 100 ms to 2 s with the rotor remaining within whisker reach through the delay period. For T5, the rotor was retracted 1.5 cm during the delay period across delays of 2 s, 3 s, and 4 s which were randomly presented with probabilities of 50%, 25%, and 25%, respectively. Fast (1.75 cm/s) and slow (0.87 cm/s) rotations of stimulus direction were used. For T1–T4, slow directions represented 5% of all trials. For T5, the fraction of slow trials was increased to 25% of all trials.

Adjustments to task settings to reinforce correct behavioral choices were carried out semi-automatically. Punishment in the form of a combination of time outs (2–10 s) and air puffs (100 ms) ranging from 1 to 5 trains to the face was manually adjusted to discourage false alarm licking on match trials. During T1, the probability of non-match stimulus conditions was manually reduced to 35–40% of all trials to reduce false alarm trials or increased up to 60% to reduce missed trials. To correct for primacy and recency stimulus bias resulting in disproportionally greater error trials for one of the two match conditions or one of the two non-match conditions, the probability of one of the two match or non-match conditions was adjusted in favor of the condition with the greater proportion of errors. Animals only received water by performing the task. Their weight was monitored daily to ensure body weight did not drop below 80% of initial weight. Animals were trained continuously and terminated once animals had performed at least 4–6 T5 sessions.

## Acetylcholine receptor inactivation
RCaMP1.07-expressing animals with Prh microprism were imaged during training through T2. Mecamylamine (1 mg/kg b.w.) or scopolamine (1–5 mg/kg b.w.) was delivered systemically via intraperitoneal (IP) injection ~1 h prior to the behavior imaging session. For control conditions, behavior imaging sessions were performed at least 16 hours after the previous pharmacological inactivation session to allow for recovery.

## Histology
Mice were anesthetized (sodium pentobarbital; 100 mg per kg and 20 mg per kg body weight) and perfused transcardially with 4% paraformaldehyde in phosphate buffer, pH 7.4. For anatomical tracing experiments, coronal sections (50–75 μm) were cut using a vibratome

## Table 3 | Training parameters to reinforce the correct choice

| Task | Goal | Criteria | Adjustment |
|---|---|---|---|
| Head-fixed | Increase punishment to correct for port bias. | Manual: >70% (hit + false alarm) | Manual: 2–10 s time out 1–10 air puffs |
| Head-fixed | Decrease punishment to reduce disengagement | Manual: 20–50% miss | Manual: |
| Home cage | Increase punishment to correct for port bias. | 50 trial sliding window <70% correct ($d'$ ~1.05) | Increase 1 s time out (10 s max) For >7 s time out, increase 1 air puff (5 max) |
| Home cage | Decrease punishment to reduce disengagement | 50 trial sliding window >50% miss | Decrease 2 s time out and 2 air puffs |
| Head-fixed | Adjust stimulus probability to correct for report biases | Manual: >70% (hit + false alarm) | Manual: Up to 0.35/0.65 (NM/M) |
| Home cage | Adjust stimulus probability to correct for report biases | 20 trial sliding window X = % trials favored port Y = % trials neglected port Moderate bias: X–Y > 0.25 Severe bias: X–Y > 0.5 | X = stim. of favored port Y = stim. of neglected : 0.35/0.65 (X/Y) severe: 0.2/0.8 (X/Y) |
| Both | Adjust stimulus probability to correct for primacy or recency stimulus bias | 20 trial sliding window For non-match stim: X = % correct fav. stim Y = % correct NM stim For match stim: X = % correct fav. stim Y = % correct M stim moderate: (X/Y–0.5) > 0.55 severe: (X/Y–0.5) > 0.6 | X = favored stim. Y = neglected stim. moderate: 0.4/0.6 (X/Y) severe: 0.3/0.7 (X/Y) |

(VT1000; Leica). For chemogenetic inactivation experiments, coronal sections (150 μm) were cut, tissue cleared and embedded in hydrogel using PACT-CLARITY, and stained for *Fos* (B4-Alexa647 hairpin amplifiers) using HCR-FISH as previously described[27]. Images were acquired using an epifluorescent microscope (Eclipse NiE, Nikon) or a spinning disk confocal microscope (Ti2-E Yokogawa Spinning Disk, Nikon).

## Two-photon imaging

Two-photon calcium imaging was performed with a custom-built resonant-scanning multi-area two-photon microscope with a 10×/0.5NA, 7.77 mm WD air objective (TL10X-2P, Thorlabs) using custom-written Scope software[31]. A 31.25 MHz 1040 nm fiber laser (Spark Lasers) was used for RCaMP1.07 imaging. Simultaneous imaging at a 32.6 Hz frame rate was performed of two imaging planes in L2/3 separated 50 μm in depth. For GRAB-Ach3.0 or GFP imaging, a single area at 32.6 Hz frame rate was acquired using an 80 MHz ti:sapphire laser (Mai Tai HP DeepSee, Spectra-Physics) tuned to 950 nm. The average power of each beam at the sample was 50–90 mW. Imaging was performed during head-fixed task behavior or during passive stimulation sessions in naïve animals using similar stimulus conditions as T5.

## In vivo image analysis

All image processing was performed in MATLAB, Python, and ImageJ as described[28,44]. For calcium imaging analysis, two-photon images were first motion corrected using a piece-wise rigid motion correction algorithm[45]. Independent noise related to photon shot noise was removed from the image times series using DeepInterpolation[46]. To identify neurons chronically imaged across all behavior sessions, a global reference image was generated by tiling FOV images from each session to account for slight variations in positioning and to reveal a common FOV shared by all sessions. ROIs were manually identified by comparing structural images based on fluorescence intensity and a map of active neurons identified by constrained non-negative matrix factorization from image time series. ROI positions were adjusted for each session to account for tissue changes or rotations over longer time scales. Calcium signals were then extracted for each ROI for each session. A global neuropil correction was performed for each neuron, and the resulting fluorescence traces were detrended on a per-trial basis. For acetylcholine imaging analysis, the fluorescence intensity across the entire FOV was averaged to obtain a bulk signal of Ach dynamics. Ach signals were *z*-scored on a per-trial basis.

## Calcium event estimation

Calcium signals were deconvolved using an Online Active Set method to Infer Spikes (OASIS), a generalization of the pool adjacent violators algorithm (PAVA) for isotonic regression[47]. First, calcium signals below baseline fluorescence (bottom 10th percentile of signal intensity) were thresholded. For each cell, a convolution kernel with exponential rise and decay time constants was determined using an autoregressive model. For measurement of photon shot noise, signal-to-noise (*v*) was calculated for each cell:

$$v = \frac{\text{Median}_t |F_{t+1} - F_t|}{\sqrt{f_r}} \qquad (1)$$

where the median absolute difference between two subsequent time points of the fluorescence trace, $F$, is divided by the square root of the frame rate, $f_r$[48]. The convolution kernel was applied to the calcium signals to obtain an initial deconvolved signal that was then normalized by the signal-to-noise resulting in a calcium event estimate ($\hat{s}$).

## Population decoding analysis

To decode population activity with respect to trial conditions, maximum margin support-vector machine (SVM) linear classifiers were used on the single-trial population response vectors of simultaneously recorded neurons within one imaging session[7]. For each neuron in the population, calcium events across a given time window were averaged for each trial and then z-scored across all trials in session time. For each classifier, activity from 10 to 20% of trials was separated for testing, while the remaining trials were used to train the classifier. In the case of comparing stimulus direction or reward, in which >100 trials were recorded for each condition (i.e., anterior vs. posterior for stimulus direction or hit vs. non-hit), the accuracy of the decoder performance was determined using 10-fold cross-validation. For comparing stimulus speed or choice in which slow speed conditions or error conditions were very few or varied across task learning (Fig. 2, Supplementary Fig. 10), trials in the minority condition in the training set were randomly resampled to match trial numbers in the other condition before 10-fold cross-validation. This process was repeated 100 times and the decoder accuracy was calculated from the average accuracy. The statistical significance of the decoding accuracy was assessed by shuffling the trial labels in the training set prior to classification. This process was repeated 1000 times, and decoder accuracies above the 95th or below the 5th percentile of the shuffled distribution were determined to be statistically significant.

For a cross-temporal classifier (Figs. 4–7, Supplementary Fig. 9), SVMs were trained as described above using average activity across the pre-stimulus period, sample period, test period, report period, or a sliding window of 1000 ms. The cross-temporal accuracy was determined using 10-fold cross-validation by testing on withheld trials from activity across different pre-stimulus periods, sample periods, test periods, report periods, or a sliding window of 300 ms. Significant cross-temporal decoding was determined by shuffling the population vector weights and then testing performance on the resulting shuffled decoder. This process was repeated 1000 times, and cross-temporal accuracies above the 95th or below the 5th percentile of the shuffled distribution were determined to be statistically significant. The decodable onset of the reward outcome classifier was defined as the first significant time point across the test and report period.

For a cross-session classifier (Fig. 4), SVMs were trained using average activity across the pre-stimulus or report period consisting of 80–90% trials from one imaging session. The cross-session accuracy was determined using 10-fold cross-validation by testing on average activity in the same trial period window in a different session using all trials. The same neuronal population imaged across sessions was used for training and testing. Significant cross-session decoding was determined by shuffling the population vector weights and then testing performance on the resulting shuffled decoder. This process was repeated 1000 times and cross-session temporal accuracies above the 95th percentile of the shuffled distribution were determined as statistically significant.

For cross-condition analysis of rewarded stimulus conditions (Fig. 5), non-match stimulus trials were separated by stimulus condition (anterior-posterior or posterior-anterior) into a training or testing set. Match stimulus trials were randomly separated into the training or testing set. SVMs were then trained using average activity from the reporting period along hit vs. non-hit trial conditions. The cross-temporal accuracy of the cross condition was determined using 10-fold cross-validation by using the average activity across a sliding window of 300 milliseconds of the test set. The cross-temporal accuracy at 300 ms from the end of the test period was used to assess the strength of the cross-condition of the test period.

## Choice selectivity

To determine the relationship between stimulus speed encoding and choice selectivity, an SVM was trained to speed trials. Neurons with significant population vector weights were determined by shuffling the trial labels in the training set prior to classification. This process was repeated 1000 times to obtain a shuffled distribution for each neuronal weight. Neuron weights above the 95th or below the 5th percentile

of the shuffled distribution were determined to be statistically significant. For significant neurons, selectivity to correct (hit, correct rejection) or error (miss, false alarm) trials was determined by calculating the average event rate for each of the two trial conditions. The peak activity level during either the sample or test period a measure of a neuron's stimulus response (SR). Choice selectivity was expressed as $(\text{SR}_{\text{ERROR}} - \text{SR}_{\text{CORRECT}})/(\text{SR}_{\text{ERROR}} + \text{SR}_{\text{CORRECT}})$, where $\text{SR}_{\text{ERROR}}$ is the peak response on error trials, and $\text{SR}_{\text{CORRECT}}$ is the peak response on correct trials.

## Computational modeling

An autoencoder was trained to reconstruct a two-dimensional input signal (Fig. 3). The input signal consisted of two independent variables, direction of movement and speed, with two different values each. This made a total of four experimental conditions: anterior direction and low speed, posterior direction and low speed, anterior direction and fast speed, and posterior direction and fast speed. These four experimental conditions were mapped to four points on a two-dimensional space [−1,−1], [−1,1], [1,−1], [1,1]. Simulations of $k$ trials per experimental condition were performed, producing a total of $4k$ trials ($k = 100$). On each trial, additive Gaussian noise with mean zero and standard deviation $\sigma_{\text{inp}}$ was added to the experimental conditions and then expanded by a random projection to an $N_{\text{inp}}$ space ($\sigma_{\text{inp}} = 0.5$, $N_{\text{inp}} = 10$).

The autoencoder consisted of input, intermediate, and output layers. Intermediate neurons were ReLU units with noise (additive Gaussian noise, $\sigma_{\text{neu}} = 1$). An additional read-out unit was included that read the intermediate layer to classify the direction of motion on a trial-by-trial basis. This additional read-out neuron was added to impose an asymmetry between the direction of motion and speed in both the intermediate and output layers. The loss function that was minimized through learning was:

$$\text{Loss} = \beta_r * \text{Loss reconstruction} + \beta_c * \text{Loss crossentropy} + \beta_s * \text{Loss sparsity}. \quad (2)$$

The reconstruction loss was the mean squared error (MSE) between the input and the output layer ($\beta_r = 0.1$). The cross-entropy loss corresponded to the classification loss of the additional read-out unit that classified the direction of motion from the activity of the intermediate layer ($\beta_c = 1$). Finally, we also added an L1-norm sparsity loss on the activity of the intermediate layer to constrain its activity ($\beta_s = 1$). The autoencoder was trained with stochastic gradient descent (ADAM, $lr = 0.002$, batch size $= 10$) for 50 epochs. A classifier (logistic regression, sci-kit learn), which can be understood as a downstream unit, was trained to read out from the familiarity population, that is, the difference between the reconstructed output and the input[34]. An independent classifier was trained on each training epoch. The reported decoding performance on both direction and speed corresponds to the mean across cross-validation iterations (5-fold CV) and independent simulations ($n = 100$).

Alternative models were trained and analyzed. This includes models containing only reconstruction loss ($\beta_r = 1$, $\beta_c = 0$, $\beta_s = 0$, $lr = 0.01$, Supplementary Fig. 8a), reconstruction and cross-entropy with respect to direction ($\beta_r = 1$, $\beta_c = 1$, $\beta_s = 0$, $lr = 0.01$, Supplementary Fig. 8b), and reconstruction, cross-entropy, and L1 sparsity on the hidden layer ($lr = 0.002$, $\beta_r = 0.1$, $\beta_c = 1$, $\beta_s = 1$, and $\beta_s = 20$, Supplementary Fig. 8c, d). Modeling was performed in Python and PyTorch. Code is available at github.com/ramonnogueira/AutoPerirhinal.

## Acetylcholine signal analysis

To understand the effects of task-relevant variables on the acetylcholine (Ach) dynamics, we fit a Normal GLM to the normalized Grab-Ach3.0 fluorescence acquired on each trial within a recording session.

The model calculates an estimated signal, $\hat{y}_t$, using:

$$\hat{y}_t = \sum_i w_i x_i(t) \quad (3)$$

where $x_i(t)$ represents the time course for the $i$th explanatory variable, and $w_i$ represents the weight assigned to this variable relating its estimated effect on the signal[49]. All GLMs were fit using MATLAB's lassoglm function with a normal distribution, identity link function, 6 penalty values ($\gamma$), and 4-fold cross-validation.

Task variables $x_i(t)$ were represented as boxcars corresponding to their occurrence during the time course of a trial. These boxcars had value "true/1" during appropriate time points and "false/0" otherwise. These include "pre-stimulus," "stimulus direction anterior," "stimulus direction posterior," and "post-trial" variables. "Reward" was represented as a boxcar lasting 300 ms after the point of reward delivery. Licking events were resampled to match the image acquisition rate. This was then convolved with a 10-sample Gaussian kernel and separated into "pre-reward licking" (Lick$_{\text{PRE}}$) and "post-Reward licking" (Lick$_{\text{POST}}$) variables based on rewarded trials. All licking on miss, false alarm, and correct rejection trials were considered Lick$_{\text{PRE}}$. For hit trials, licks before the water reward were Lick$_{\text{PRE}}$, while licks after water reward were Lick$_{\text{POST}}$.

Related covariates were grouped together into 'task factors.' Each task variable was treated as its own "task factor" with the exception of "stimulus direction anterior" and "stimulus direction posterior" which were grouped into a task factor for "stimulus direction." For each task factor, a partial model was constructed that excluded the covariates associated with this task factor. Any increase in deviance from the full model to the partial model therefore resulted from the exclusion of this task factor's covariates. Akaike Information Criterion (AIC) was used to compare deviance between partial models in which different numbers of covariates were excluded such that:

$$\text{AIC} = 2k - 2\ln(L) = 2k + \text{deviance} \quad (4)$$

where $k$ is the number of model parameters, deviance $= -2\ln l$, and $L$ is the model likelihood. The difference in AIC ($\Delta$AIC) between the full and partial model was calculated as:

$$\Delta\text{AIC} = \text{AIC}_{\text{partial}} - \text{AIC}_{\text{full}} \quad (5)$$

## Statistical procedures

No statistical methods were used to predetermine the sample size. For Prh inactivation experiments, investigators were blinded to hM4Di+ or hM4Di− groups during experiments and outcome assessment. For two-photon experiments, animals were not randomized, and the investigators were not blinded to allocation during experiments and outcome assessment. Statistical tests used are indicated in figure legends. Error bars on plots indicate standard error of the mean (SEM) unless otherwise noted.

For Prh inactivation experiments, a bootstrap analysis was used to compare the fraction hM4Di+ versus hM4Di− animals able to successfully accomplish the T2 stage. For testing of sequence reliability or stimulus similarity across passive and training stages, a one-way ANOVA was performed, followed by a multiple comparisons test. For testing of differences in linear decoder or cross-temporal decoder performance in individual sessions between training stages, a one-way ANOVA was performed, followed by a multiple comparisons test. For the performance of linear decoders for direction or speed, a Student's $t$-test was used to compare correct versus error trials at specific training stages. For comparisons of choice selectivity in individual neurons across training stages, a one-way ANOVA was performed, followed by a multiple comparisons test. For statistical tests of Ach

signal encoding, a repeated-measures ANOVA was performed, followed by a multiple comparisons test used to compare the strength of GLM ΔAIC values between task factors. A Student's $t$-test was used to compare AP versus PA decoder performance as well as cross-conditional decoder performance at specific T3 sessions. The Bonferroni-Holm method was used to correct for multiple comparisons.

## Reporting summary

Further information on research design is available in the Nature Portfolio Reporting Summary linked to this article.

## Data availability

Source data in this study are available in a G-Node GIN repository [https://doi.org/10.12751/g-node.m36s4g][50]. Source data are provided with this paper.

## Code availability

Generalized linear model code is available at github.com/common-chenlab. Autoencoder model code is available at github.com/ramon-nogueira/AutoPerirhinal [https://doi.org/10.5281/zenodo.10581525][51]. Custom Scope software used for data collection is available is available at http://rkscope.sourceforge.net.

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

## Acknowledgements

We thank Y. Livneh for guidance in prism implant surgeries, A. Dong, F. Wang, A Deng for assistance in home cage training system design, A. Williams, M.W. Howard for guidance in data analysis, M. Hasselmo for comments on the manuscript. This work was supported by grants from the Richard and Susan Smith Family Foundation (J.L.C.), Elizabeth and Stuart Pratt Career Development Award (J.L.C.), Whitehall Foundation (J.L.C.), Harvard NeuroDiscovery Center (J.L.C.), Boston University Kilachand Fund Award (J.L.C.), National Institutes of Health BRAIN Initiative Award R01NS109965 (J.L.C.), National Institutes of Health New Innovator Award DP2NS111134 (J.L.C.). NSF Neuronex 1707398 (S.F.); the Gatsby Charitable Foundation GAT3708 (S.F.), the Simons Foundation (S.F.), the Swartz Foundation (S.F. and R.N.).

## Author contributions

D.G.L., C.A.M. and J.L.C. designed the study. D.G.L. and G.H. designed the home cage training system, D.G.L., A.E.C. and G.H. performed home cage inactivation experiments, and D.G.L., C.A.M. and O.K. performed two-photon imaging experiments. R.N. performed computational modeling. G.D.L. performed retrograde tracing experiments. D.G.L., D.L.M. and J.L.C. performed data analysis. R.N., S.F. and J.L.C. supervised data analysis. D.G.L., C.A.M. and J.L.C. wrote the paper.

## Competing interests

The authors declare no competing interests.
