## [Peer Review File · Nature Communications]

Perirhinal cortex learns a predictive map of the task environmentEditorial Note: This manuscript has been previously reviewed at another journal that is not operating a transparent peer review scheme. This document only contains reviewer comments and rebuttal letters for versions considered at Nature Communications.

REVIEWER COMMENTS

Reviewer #1 (Remarks to the Author):

Overall, the revision is somewhat improved again, albeit without a strong effort to directly address the concerns fully.

The authors moved the freely moving behavior to the supplement. It's true that this behavior, which differs in important ways from the head-fixed behavior, may nevertheless be of interest to some readers. However, if included in the supplement, then the issues that I raised in the previous round of review should each be discussed in the Supp Figure legend and Methods. Moving data to the supplement does not mean that interpretability issues needn't be addressed, particularly when the claim in main text remains similar. (Please also check for typos in revised text and throughout – while these can happen to anyone, they further decrease reviewer confidence in the care taken throughout.)

Regarding the use of the word 'mediated' to describe cholinergic signaling. Reasonable people can disagree, but to me, mediate is not something to use unless something is a main mechanism, not just if it's involved somehow. Particularly for use in the abstract. I find it confusing why this wasn't just modified as requested. It would seem in the authors' best interest in terms of future reader evaluations of the paper to heed advice regarding matching the wording to the partial nature of the result.

In regards to my comments on Figures 7-8, again some of the first questions were addressed while the later ones regarding use of different decoders were not. Why not just answer all reviewer questions one by one?

Regarding the statistical concerns about not doing statistics across mice: The authors provide some partial addressing of this point, but perhaps didn't think others would have this concern? At least they didn't indicate in the rebuttal where they added the requested analyses. The main claims will benefit from demonstrating mouse-by-mouse rigorous statistics, at least in the supplement, and throughout the paper for all main conclusions.

Reviewer #3 (Remarks to the Author):

I had relatively minor comments in the last round of review, and the authors did a good job addressing them. I'm satisfied.

Response to Reviewer's Common Concerns (reviewer's comments in regular font; our comments in bold)

Reviewer #1 (Remarks to the Author):

Overall, the revision is somewhat improved again, albeit without a strong effort to directly address the concerns fully.

The authors moved the freely moving behavior to the supplement. It's true that this behavior, which differs in important ways from the head-fixed behavior, may nevertheless be of interest to some readers. However, if included in the supplement, then the issues that I raised in the previous round of review should each be discussed in the Supp Figure legend and Methods. Moving data to the supplement does not mean that interpretability issues needn't be addressed, particularly when the claim in main text remains similar. (Please also check for typos in revised text and throughout – while these can happen to anyone, they further decrease reviewer confidence in the care taken throughout.)

It's not clear whether or not the reviewer saw our supplementary text in yellow starting in line 1187. We have added a few more sentences to address the issues at line 1208. We think it is more appropriate that these issues are discussed here rather than the supp figure legend and methods.

Regarding the use of the word 'mediated' to describe cholinergic signaling. Reasonable people can disagree, but to me, mediate is not something to use unless something is a main mechanism, not just if it's involved somehow. Particularly for use in the abstract. I find it confusing why this wasn't just modified as requested. It would seem in the authors' best interest in terms of future reader evaluations of the paper to heed advice regarding matching the wording to the partial nature of the result.

I think we have to agree to disagree. There are different words for different degrees of involvement. We should that cholinergic signaling is necessary for both behavior and relevant activity patterns in Prh. This is sufficient by our standards to use the word 'mediate'. Whether it is the main mechanism, would require testing 20-30 other plausible mechanisms and ruling them out one by one. This is hardly ever done in systems neuroscience and yet claims of mechanism can be quite liberal. We believe our choice of words is reasonable for the field.

In regards to my comments on Figures 7-8, again some of the first questions were addressed while the later ones regarding use of different decoders were not. Why not just answer all reviewer questions one by one?

The way the reviewer phrased their questions suggested to us that the reviewer was throwing out ideas as opposed to demanding that each analysis be run. Some of the analyses requested such as the PCA do not make much sense with respect to the original analysis, so it was unclear why the reviewer thought it was necessary. Since we removed the original analysis that raised the concern from the manuscript, these suggestions for additional analysis are no

longer relevant. There is not a need to perform confirmatory analysis on a prior analysis that no longer exists in the manuscript.

Regarding the statistical concerns about not doing statistics across mice: The authors provide some partial addressing of this point, but perhaps didn't think others would have this concern? At least they didn't indicate in the rebuttal where they added the requested analyses. The main claims will benefit from demonstrating mouse-by-mouse rigorous statistics, at least in the supplement, and throughout the paper for all main conclusions.

The reviewer had the opportunity to raise this concern in their first round of review but waited until the 3rd round of review to request this change. In the prior response, we had provided some analysis to assuage the reviewer that this is not an issue, but to redo the analysis for the entire paper at such a late stage seems a little egregious. The data and code will be made publicly available and so readers will have the opportunity to re-run the analysis as they would like.